# Physiologic recovery of *Mycobacterium tuberculosis* from drug injury: A molecular study of post antibiotic effect *in vitro* and *in vivo*

Jo Hendrix[1,2,3], Reem Al Mubarak[1,2,3], Matthew J. Reichlen[3,4], Samuel T. Tabor[1,2,3,4], Adeline Bateman[1,2,3], Lisa M. Massoudi[5], Karen Rossmassler[2,3,6], Firat Kaya[7], Matthew D. Zimmerman[7], Holly Nielsen[1,2,3], Elizabeth A. Wynn[2,3,8], Martin I. Voskuil[3,4], Gregory T. Robertson[3,5], Camille M. Moore[3,8,9], Nicholas D. Walter[1,2,3]*

1 Division of Pulmonary, Allergy and Critical Care Medicine, University of Colorado Anschutz Medical Campus, Aurora, Colorado, United states of America, 2 Rocky Mountain Regional VA Medical Center, Aurora, Colorado, United states of America, 3 Consortium for Applied Microbial Metrics, Aurora, Colorado, United states of America, 4 Department of Immunology and Microbiology, University of Colorado Anschutz Medical Campus, Aurora, Colorado, United states of America, 5 Mycobacteria Research Laboratories, Department of Microbiology, Immunology and Pathology, Colorado State University, Fort Collins, Colorado, United states of America, 6 Linda Crnic Institute for Down Syndrome, University of Colorado Anschutz Medical Campus, Aurora, Colorado, United states of America, 7 Center for Discovery and Innovation and Hackensack School of Medicine, Hackensack Meridian Health, Nutley, New Jersey, United states of America, 8 Center for Genes, Environment and Health, National Jewish Health, Denver, Colorado, United states of America, 9 Department of Biostatistics and Informatics, University of Colorado Anschutz Medical Campus, Aurora, Colorado, United states of America

* Nicholas.Walter@cuanschutz.edu

## Abstract

Post-antibiotic effect (PAE) describes the delay in bacterial growth that continues after antibiotics are cleared. The physiologic basis of PAE in Mycobacterium tuberculosis (*Mtb*) remains poorly understood. Here, we evaluated the long-standing hypothesis that PAE reflects the time required for bacteria to recover from drug-induced physiologic damage by comparing *Mtb* after varying durations of treatment with the four-drug isoniazid, rifampin, pyrazinamide, ethambutol combination *in vitro* and in BALB/c mice using two novel molecular readouts of bacterial health. In aerobic axenic culture and in the high-dose aerosol mouse model, quantification of *Mtb* rRNA synthesis via the RS ratio and *Mtb* transcriptional profiling via SEARCH-TB revealed that longer drug exposure was associated with greater injury and adaptation during treatment, as well as slower recovery after treatment, *i.e.,* longer PAE. Recovery followed a conserved sequence, from resumption of rRNA synthesis, to broad transcriptional reprogramming, to eventual CFU change. In mice, however, physiologic recovery was markedly slower and less complete than *in vitro*, indicating longer PAE in the context of immunity. Our observation that PAE in *Mtb* depends on the duration of drug exposure and correlates with the degree of bacterial injury support the hypothesis that nonlethal physiologic damage contributes to PAE. Our observation that PAE of the standard TB regimen is longer in mice than *in vitro* indicates that

**Data availability statement:** Murine sequence data are available on NCBI via BioProject PRJNA1140268 under the title "SEARCH-TB of Mtb after HRZE Treatment Interruption in the BALB/c Mouse." The in vitro sequence data are available on NCBI via BioProject PRJNA1425933, under the title "Physiologic recovery of Mycobacterium tuberculosis from drug injury: A molecular study of post antibiotic effect in vitro." To make the data more easily accessible to the research community, we have also created an interactive data analysis tool available at: https://microbialmetrics.org/analysis-tools/.

**Funding:** NDW acknowledges funding from Veterans Affairs 1I01BX004527-01A1. GR and NDW acknowledge funding from Bill and Melinda Gates Foundation OPP1170003. GR and NDW acknowledge funding from NIH UM1 AI179699. The funders had no role in study design, data collection and analysis, decision to publish, or preparation of the manuscript.

**Competing interests:** The authors have declared that no competing interests exist.

immunity augments PAE for *Mtb*. Molecular evaluation of bacterial physiology provides a new basis for probing recovery from drug exposure and understanding PAE.

## Author summary

Tuberculosis (TB) is the leading cause of death from infection worldwide. Antibiotics used to treat *Mycobacterium tuberculosis* (*Mtb*) often inhibit bacterial growth well beyond the time that drugs are cleared and unmeasurable. It has long been hypothesized that this post-antibiotic effect (PAE) could reflect time required for bacteria to repair sub-lethal cellular damage. A barrier to understanding is that PAE has traditionally been studied by enumerating bacterial burden rather than evaluating change in bacterial health. Additionally, the PAE of TB drugs has been studied exclusively *in vitro* rather than in the context of immunity. Here, we used several novel molecular measures of bacterial health in *in vitro* and mouse experiments to evaluate *Mtb* recovery after short treatment with the global standard 4-drug combination treatment. We discovered three novel findings. First, longer treatment resulted in longer PAE. Longer treatment caused progressively greater injury and adaptation during treatment and slower physiologic recovery afterwards, suggesting that the worse the damage, the slower the recovery. Second, recovery followed a reproducible sequence, beginning with restoration of rRNA synthesis, followed by widespread transcriptional reprogramming, and finally increases in bacterial numbers. Third, although this recovery sequence was conserved between *in vitro* and mice, recovery in mice was markedly slower and incomplete, resulting in a substantially longer PAE. Together, these findings support the damage hypothesis as a component of PAE and highlight the importance of immunity in controlling *Mtb* after drug treatment.

## Introduction

Tuberculosis (TB) remains the leading cause of death from infection worldwide. There is an urgent need for new treatments that can cure TB more rapidly and reliably [1]. TB drugs exert post-antibiotic effect (PAE), meaning *Mycobacterium tuberculosis* (*Mtb*) is slow to recover and resume growth following antibiotic exposure [2,3]. PAE is thought to be clinically relevant as it may underly the degree to which a regimen retains efficacy even when patient adherence is sub-optimal [4,5].

One decades-old hypothesis is that PAE occurs because bacteria require time to recover from antibiotic-induced damage [6,7] Engagement of an antibiotic with its target molecule initiates a secondary injury cascade of drug-specific physiologic perturbations [8–10],that can cause macromolecular damage such as DNA breaks [11] or ribosomal degeneration [12]. PAE has classically been defined as change in bacterial burden using measures such as colony forming units (CFU) [2,6,13,14], optical density (OD$_{600}$) [15–17], or time-to-detection in Bactec culture [2,18,19] However,

enumeration of bacterial burden provides no information about change in *Mtb* physiology [20], the hypothesized basis of PAE. There is a paucity of information about bacterial physiology as *Mtb* recovers during the PAE period.

Here, we evaluated PAE in a novel way by evaluating how *Mtb* physiology recovers after exposure to the global standard isoniazid, rifampin, pyrazinamide and ethambutol (HRZE) regimen. We used new molecular measures of bacterial physiology including the RS ratio and SEARCH-TB. The RS ratio quantifies *Mtb* rRNA synthesis based on detection of short-lived precursor rRNA sequence [20]. A higher RS ratio indicates an active *Mtb* phenotype with higher ongoing rRNA synthesis whereas a low RS ratio indicates a quiescent *Mtb* phenotype with less ongoing rRNA synthesis. Mouse studies have shown an association between the degree to which regimens suppress the RS ratio and time to non-relapsing cure [20]. We also used SEARCH-TB, a highly sensitive *Mtb*-targeted RNA-seq platform, to evaluate the entire *Mtb* transcriptome.[8] As previously demonstrated [8], SEARCH-TB is substantially more sensitive than qPCR and standard RNA-seq methods, providing a highly granular measure of *Mtb* gene transcription. These tools enabled us to quantify both the physiologic injury and adaptation that occurs during drug exposure and the recovery that occurs after drug exposure. Collectively, the RS ratio and SEARCH-TB have been described as measures of "pathogen health" to contrast with more traditional measures of pathogen burden.

A second important gap in existing research is that PAE for TB drugs has not been evaluated *in vivo*. For other bacteria, the *in vivo* environment and presence of immunity are known to influence PAE [21,22] making PAE substantially longer *in vivo* than *in vitro*. To our knowledge, *in vivo* evaluation of PAE for TB drugs has been limited to a predictive model that suggests PAE may be twice as long *in vivo* [23].

Here, we evaluated the PAE of the standard HRZE regimen both *in vitro* and in mice using not only traditional assays of bacterial burden but also molecular measures of pathogen health. We found that longer antibiotic exposure caused progressively greater bacterial injury and adaptation, resulting in slower and less complete recovery after treatment cessation. Recovery followed a conserved and ordered sequence, beginning with restoration of rRNA synthesis, followed by transcriptional reprogramming, and ultimately increases in bacterial burden. Although this sequence was preserved across experimental systems, recovery in mice was markedly slower and more constrained than *in vitro*, indicating that bacterial recovery within a host is shaped by immune and environmental pressures absent in culture. Together, these findings support the hypothesis that physiologic damage is an important contributor to PAE.

## Methods

### Ethics statement

All animal procedures were approved by the Colorado State University Animal Care and Use Committee (Reference number: 1515) and conducted according to established guidelines.

### *in vitro* experimental design

For *in vitro* PAE experiments, *Mtb* Erdman cells were inoculated into 7H9 media and grown to mid-log. At mid-log, starter cultures were split into separate sterile 125 ml Erlenmeyer flasks, one each for no drug control, 6-hour and 48-hour drug exposure arms each adjusted to an $OD_{600}$ of 0.05 in 50 ml fresh 7H9. Cultures were then outgrown for 18 h at 37$^\circ$C prior to drug addition. For drug exposed cultures, a 50 µl drug suspension was added from a 1000X sterile stock solution containing the following: 0.05 mg/ml isoniazid (Sigma-Aldrich), 0.05 mg/ml rifampicin (Chem-Impex-Int'l), 1.6 mg/ml pyrazinamide (MP Biomedicals), 2.0 mg/ml ethambutol (MP Biomedicals). Drugs were suspended in DMSO at the indicated concentrations, sterilized using a 0.22 µm nylon syringe filter, and stored at -80$^\circ$C prior to use. Control non-drug exposed cultures were collected by centrifugation for 5 min at 3220 x *g*, washed 2 times with 50 ml 7H9 and then resuspended in 50 ml fresh 7H9 to an $OD_{600}$ of 0.05. Cultures were returned to the incubator and grown at 37$^\circ$C for the pre-defined drug exposure period. At the end of the defined drug exposure period, cultures were collected by centrifugation, washed 2 times with 50 ml 7H9 to remove residual drug and then resuspended in 50 ml fresh 7H9 to an $OD_{600}$ of 0.05.

Following drug wash out, cultures were returned to the incubator and grown at 37°C until cultures reached an $OD_{600}$ of 2.0 or exceeded 15 days. Samples for $OD_{600}$ and CFU were collected immediately prior to drug addition, at the end of the drug exposure period immediately prior to drug wash out, immediately post washing, and once daily post washing until reaching the growth endpoint (**Fig A in** S1 File). RNA was collected daily for RS ratio except in the 48-hour arm where RNA was only collected every other day after day 9. SEARCH-TB was performed on days indicated in **Fig 2D**. RNA collection and isolation and CFU plating and enumeration were performed as previously described [24].

## Calculation of PAE duration

For each treatment group, CFU data collected immediately before and after crossing the normalized threshold $y = 1$ were used to estimate the time at which this threshold was reached. Median CFU values were computed for each group and washout period. The time at which $y = 1$ occurred was then calculated by linear interpolation between the last timepoint below the threshold and the first timepoint above it, using the formula:

$$t_{y=1} = t_1 + \frac{(1 - y_1)(t_2 - t_1)}{(y_2 - y_1)}$$

where $(t_1, y_1)$ is the timepoint immediately before the threshold, and $(t_2, y_2)$ is the timepoint immediately after.

The post-antibiotic effect (PAE) time was defined as the delay, relative to the no-drug control, in reaching the $y = 1$ threshold.

**Murine experiments.** We used the BALB/c high-dose aerosol infection model, a mainstay of contemporary TB drug development [13]. Female BALB/c mice, 6–8 weeks old, were exposed to aerosol (Glas-Col) with *Mtb* Erdman, resulting in 4.43 $\log_{10}$ CFU in lungs on day one. On infection day 11, five mice were euthanized as pre-treatment controls, and HRZE treatment was initiated via oral gavage five days a week. We used established standard doses of all antibiotics [25] (**Table A in** S1 File). End of treatment mice (N = 4 per treatment duration) were sacrificed on days 12 (2-weeks) and 26 (4-weeks), one day after their final HRZE dose. Additional mice (N = 4 per group) were sacrificed on days 4 or 5, 7, 11, 14, 21, and 28 after treatment interruption. Lungs were flash frozen before CFU enumeration and RNA extraction as recently described [20].

## RNA extraction, molecular profiling, and data preparation

Lung tissue was thawed in a 4M guanidine thiocyanate- Tris(2-carboxyethyl)phosphine buffer before mechanical tissue disruption and *Mtb* lysis via beadbeating. As previously described [20,26] RNA was extracted using the simply RNA tissue kit (AS1340), on the Maxwell RSC 48 (Promega) instrument. The RS ratio was quantified using published primer/probes for ETS1 and 23S via digital PCR (BioRad QX200 system). Sequence analysis of samples was performed via SEARCH-TB following recently described methods [8]. Briefly, RNA was reverse transcribed and cDNA targets were then amplified using the SEARCH-TB panel, following the Illumina AmpliSeq protocol. Libraries were sequenced on an Illumina NovaSeq6000. We followed the bioinformatic analysis and quality control pipeline as described in Supplemental Methods.

**In vitro minimum inhibitory concentration testing.** The Minimal Inhibitory Concentration (MIC) was determined for pyrazinamide, rifampicin, ethambutol, or isoniazid against *Mtb* Erdman in 7H9 media supplemented with 0.2% [v:v] glycerol and 10% [v:v] ADC, with 0.05% [v:v] Tween-80 (7H9 media), pH 6.6. MIC values were determined by a broth microdilution assay using two-fold serial drug dilutions. The lowest consecutive antimicrobial concentration that showed a ≥ 80% reduction in $OD_{600}$ relative to drug-free control wells, was regarded as the MIC.

**Pharmacokinetic measurements.** After euthanizing mice individually, blood was collected via cardiac puncture into K3EDTA tubes (Greiner Bio-One MiniCollect 1 mL K3E K3EDTA item#450474), maintained on ice before centrifugation at 10,000 RCF for 2 min at 4°C, plasma was transferred, then frozen at -80°C within 1 hour of collection. Drug levels in

plasma were quantified by high pressure liquid chromatography coupled to tandem mass spectrometry (LC-MS/MS) using previously documented methods for isoniazid [27], rifampicin [27], pyrazinamide [28], and ethambutol [29].

**Transcriptome comparisons.** We evaluated the effect of HRZE treatment *in vitro* by comparing gene expression at the end of 6-hour or 48-hour treatment to day 0 untreated controls. We evaluated recovery by comparing: (1) the end of recovery (as defined as the samples having increased by 1.5 log CFU since the wash-out) to the end of treatment and (2) the end of recovery in the 48-hour arm to the end of recovery in the 6-hour arm.

We evaluated the effect of HRZE treatment *in vivo* by comparing gene expression at the end of 2-week or 4-week treatment with pre-treatment controls. We evaluated recovery by comparing: (1) the end of 28-day recovery with the end of 2-week or 4-week treatment, (2) the end of 28-day recovery with pre-treatment controls and (3) the end of 28-day recovery following 4-week treatment with the end of 28-day recovery following 2-week treatment.

For each gene, negative binomial generalized linear models were fit using edgeR [30,31] to identify the effect of treatment and recovery. For each comparison of interest, likelihood ratio tests were performed and genes with Benjamini–Hochberg adjusted P-value [32] less than 0.05 were considered significantly differentially expressed.

**Bacterial average expression comparison.** The average normalized gene expression for genes in categories established by Cole et al [33].or curated from the literature (**Table B in** S1 File; S2 File) were calculated for each sample as previously described [8]. Values were compared between pairs of timepoints using a t-test. Gene categories with Benjamini–Hochberg adjusted P-values [34] less than 0.05 were considered significant. Differential expression, functional enrichment, and visualizations can be evaluated interactively using an Online Analysis Tool created using the R package Shiny [35] v1.8.1. [https://microbialmetrics.org/analysis-tools/]

## Results

### Effect of HRZE in aerobic axenic culture

Here, we describe the *in vitro* effect of 6-hour and 48-hour HRZE exposure (**Fig 1a**) on bacterial burden and bacterial health before describing recovery from drug exposure below. CFU was unchanged after 6-hour HRZE exposure and declined 74.7% (0.6 log; $p = 0.001$) after 48-hour exposure (**Fig 1b**). HRZE reduced the median RS ratio from 304 (a level consistent with a high rate of ongoing rRNA synthesis) to 18.7 after 6-hour and and 0.8 after 48-hour exposure (indicating marked suppression of rRNA synthesis) (**Fig 1c**). HRZE significantly altered expression of 2,078 (58.5%) and 2,845 (80.1%) genes relative to pre-treatment control, respectively (Fig 1d-1e). Juxtaposing fold-change after 6-hour and 48-hour exposures relative to control (**Fig 1f**) showed a "shift" away from the identity line with larger fold-changes after longer treatment, indicating a more extreme change with longer exposure. The direction of transcriptional change was generally concordant between 6-hour and 48-hour exposures. In hierarchical clustering, Clusters 1 (N = 566 genes) and 2 (N = 951) decreased over time while Clusters 3 (N = 760 genes) and 4 (N = 858 genes) increased over time (**Fig 1g**). Gene set enrichment analysis for Clusters 1 and 2 combined (**Fig 1h**) showed that HRZE progressively suppressed genes associated with growth, transcription, translation, synthesis of rRNA proteins, and immunogenic secretory peptides. Gene set enrichment analysis for Clusters 3 and 4 combined (**Fig 1i**) showed HRZE progressively induced expression of the DosR regulon and insertion sequences and phage-related genes (Full enrichment results in **Table C in** S1 File).

### Post-antibiotic effect in aerobic axenic culture

OD and CFU (Fig 2a-2b) showed a lag in *Mtb* growth that increased with longer exposure to HRZE. After 6-hour and 48-hour HRZE exposure, PAE was 60.9 hours and 174 hours, respectively.

### Physiologic recovery after drug exposure in aerobic axenic culture

During recovery after 6-hour and 48-hour HRZE exposure, the RS ratio rose significantly at the earliest timepoints evaluated ($p < 0.0001$ relative to end of treatment). The rate of increase was slower in the 48-hour arm. The 6-hour and 48-hour

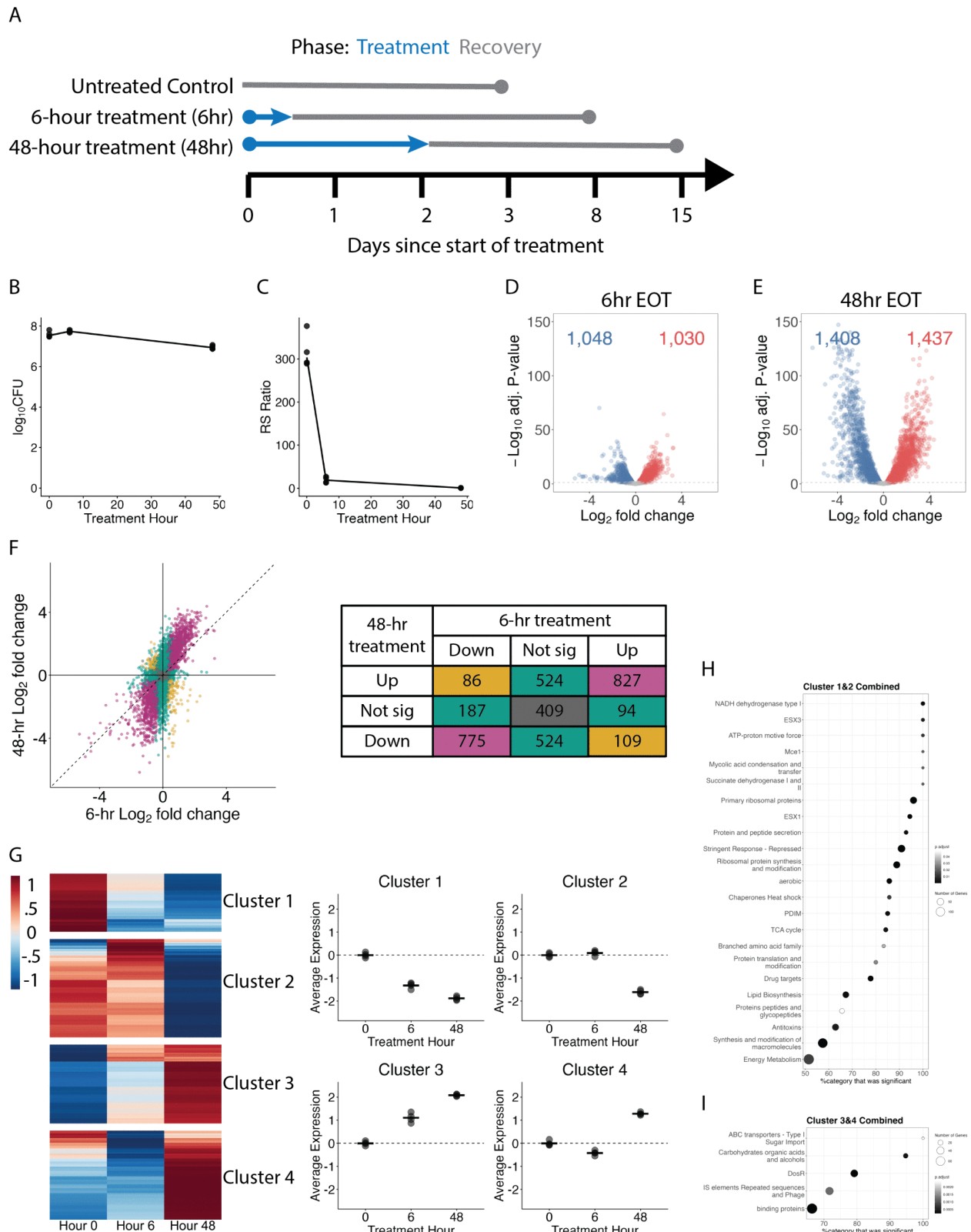

**Fig 1. Effect of treatment in axenic culture. (a)** Timeline of treatment for experimental arms for *in vitro* study. Cultures were untreated, treated for 6 hours, or treated for 48 hours. Cultures underwent washout to remove drugs, diluted to an OD of 0.05, then allowed to grow until exceeding an $OD_{600}$ of

2 or 15 days of recovery. **(b)** CFU burden after 0, 6, and 48 hours of treatment. Each dot represents one sample. The line connects the median values. **(c)** RS ratio as displayed in **(b)**. **(d-e)** Volcano plot showing $\log_2$ fold changes and $-\log_{10}$ P-values of genes differentially expressed after (d) 6-hour or (e) 48-hour treatment compared to pre-treatment control. Genes significantly down- and up-regulated relative to control (adj. P<0.05) are shown in blue and red, respectively. **(f)** Log fold change of genes after 48-hour treatment compared to pre-treatment control versus 6-hour treatment compared to pre-treatment control. Purple shading indicates genes with concordant fold-change direction and significance between treatment times. Teal shading indicates genes that were significantly differentially expressed under one treatment duration but not both. Gold shading indicates genes that were significant for both treatment durations but in opposite directions. Gray shading indicates genes that were not significantly differentially expressed in either study. Table specifies the number of genes per category. **(g)** Hierarchical clustering of genes differentially expressed over time with HRZE treatment in axenic culture (N=3,135 genes). The heatmap shows the batch-adjusted, VST-normalized, scaled gene expression averaged across samples. Clustering identified four broad patterns. Dot plots show the average expression for each cluster across time points. Each point represents an individual sample. Horizontal lines indicate average values. Values are centered around the average value for the pre-treated samples so that points above and below zero represent upregulation and downregulation relative to pre-treatment, respectively. **(h-i)** Dot plots indicate *Mtb* gene categories that were enriched in (h) clusters 1 and 2 or (i) clusters 3 and 4. Dot size indicates the number of genes from a category that were present in the clusters and x-axis indicates what percent of the category was present. Coloration indicates significance. Only the top 10 enrichments are shown.

exposure arms required approximately 2 and 6 days, respectively, to recover to RS ratio values of the untreated control (**Fig 2c**).

Use of SEARCH-TB at intervals during recovery (**Fig 2d**) illustrate the kinetics of transcriptional change during recovery. Heatmaps with clustering over time (**Fig 2e**), revealed slower transcriptional recovery in the 48-hour exposure arm than the 6-hour arm. In the 6-hour arm, the small Cluster 3 (N=368 genes) changed during the first 1.75 days of recovery and by 3.75 days the larger gene Clusters 1 (N=1,665) and 4 (N=1,474) also changed. By contrast, in the 48-hour arm, there appeared to be a period of transcriptional stasis. For example, Clusters 2 and 4 remained unchanged 4 days after the end of drug exposure. A change in these clusters was discernable at the next recovery timepoint tested (*i.e.,* on day 8).

## Recovery of specific *Mtb* physiologic processes in aerobic axenic culture

This section describes change in gene categories summarized in **Table B in** S1 File. Expression change can be further explored interactively via our Online Analysis Tool. (https://microbialmetrics.org/analysis-tools/).

**Metabolism-related genes *in vitro*.** Both 6-hour and 48-hour HRZE exposure decreased expression of metabolism related genes (**Fig 3a**), but this effect was significantly greater for the 48-hour exposure as indicated by (1) TCA cycle genes (*Adj-P<0.00001* after 48-hour relative to 6-hour exposure), (2) NADH dehydrogenase type I genes (*Adj-P<0.00001*), (3) NADH dehydrogenase type II genes (*Adj-P<0.001*), (4) succinate dehydrogenase types I and II (*Adj-P<0.00001*) and (5) ATP synthetase genes (*Adj-P<0.0001*). A transition in oxidative phosphorylation was signaled by a shift in utilization of cytochrome oxidases. Specifically, both 6-hour and 48-hour HRZE exposures decreased expression of the primary cytochrome *bcc*/*aa3* supercomplex (**Fig 3b**) but this effect was significantly greater for the 48-hour exposure (*Adj-P<0.00001*). By contrast, HRZE induced expression of the less-efficient alternative cytochrome *bd* oxidase at 48-hours but not at 6-hours (**Fig 3c**). The above changes in metabolism-related genes were entirely reversed during recovery but the 48-hour arm took longer than the 6-hour arm (**Fig B in** S1 File).

**Protein synthesis-related genes *in vitro*.** Both 6-hour and 48-hour HRZE exposure decreased expression of protein synthesis genes, but this effect was significantly greater for the 48-hour exposure as indicated by protein translation and modification genes (*Adj-P<0.00001* after 48-hour relative to 6-hour exposure; **Fig 3d**) and primary ribosomal protein genes (*Adj-P<0.00001*; **Fig B in** S1 File). These changes were entirely reversed during recovery but the recovery took longer in the 48-hour arm.

**Cell wall-associated genes *in vitro*.** Both 6-hour and 48-hour HRZE exposure decreased expression of cell-wall associated genes (**Fig 3e**; **Fig B in** S1 File), but this decrease was greater for the 48-hour exposure as indicated by Antigen 85 (*Adj-P<0.00001* after 48-hour relative to 6-hour exposure), arabinogalactan synthesis (*Adj-P<0.01*), mycolic

PLOS Pathogens

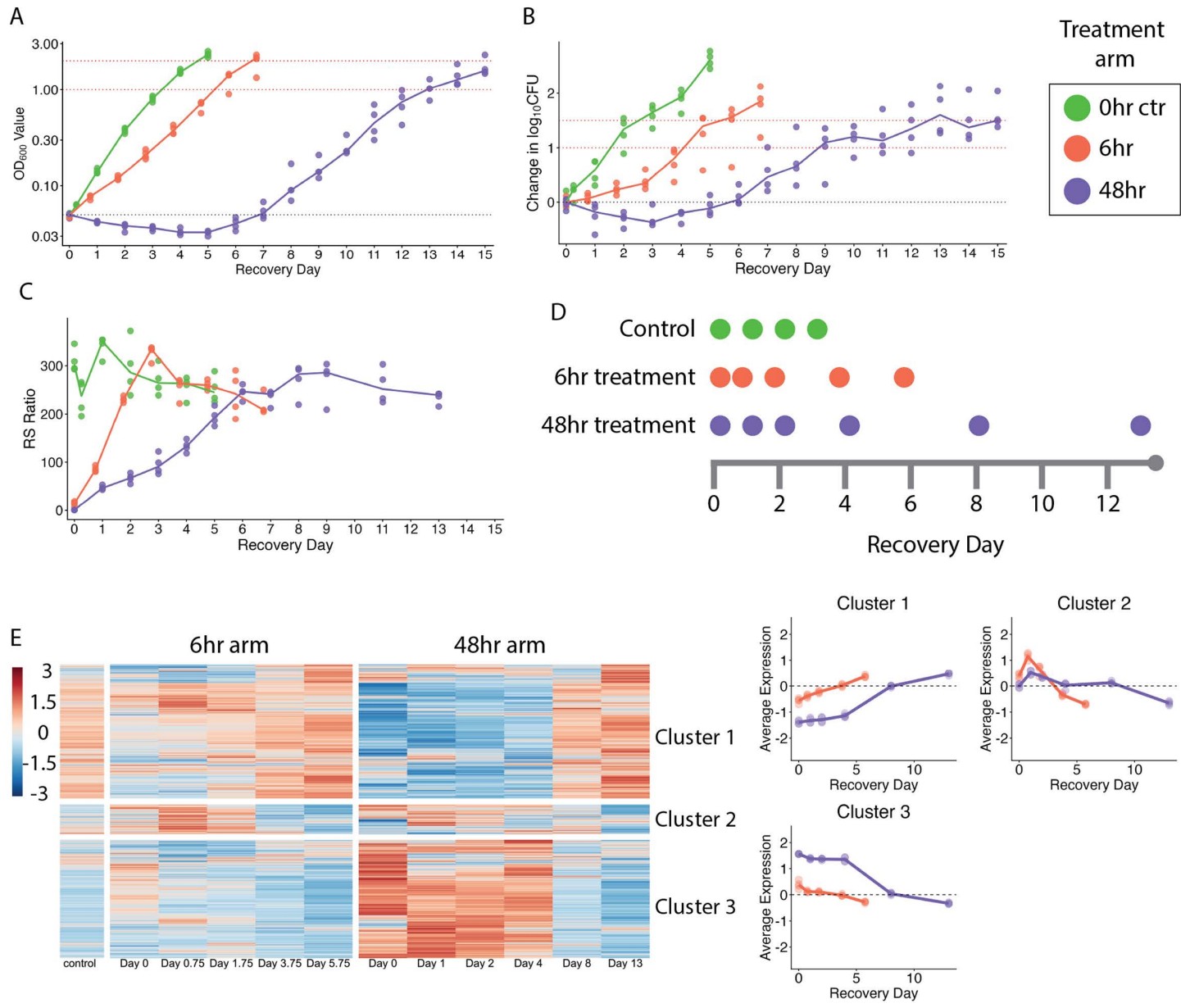

**Fig 2. Overview of recovery in axenic culture. (a)** OD$_{600}$, **(b)** change in CFU, and **(c)** RS ratio of samples during the PAE phase of recovery after being diluted to OD$_{600}$ of 0.05 and allowed to grow in the absence of HRZE. Each dot represents one sample, and lines connect the median values across timepoints for untreated control (green), 6-hour treatment (orange), and 48-hour treatment (purple) group. **(b)** Timeline for sampling each *in vitro* treatment group for SEARCH-TB sequencing and analysis. **(e)** Hierarchical clustering of genes differentially expressed during PAE phase (N = 3,507 genes). The heatmap shows the batch-adjusted, VST-normalized, scaled gene expression averaged across samples. Clustering identified four broad patterns. Dot plots show the average expression for each cluster across time points. Each point represents an individual sample from 6-hour treatment (orange) or 48-hour treatment (purple) groups. Horizontal lines indicate average values. Values are centered around the average value for the pre-treated samples so that points above and below zero represent upregulation and downregulation relative to pre-treatment, respectively.

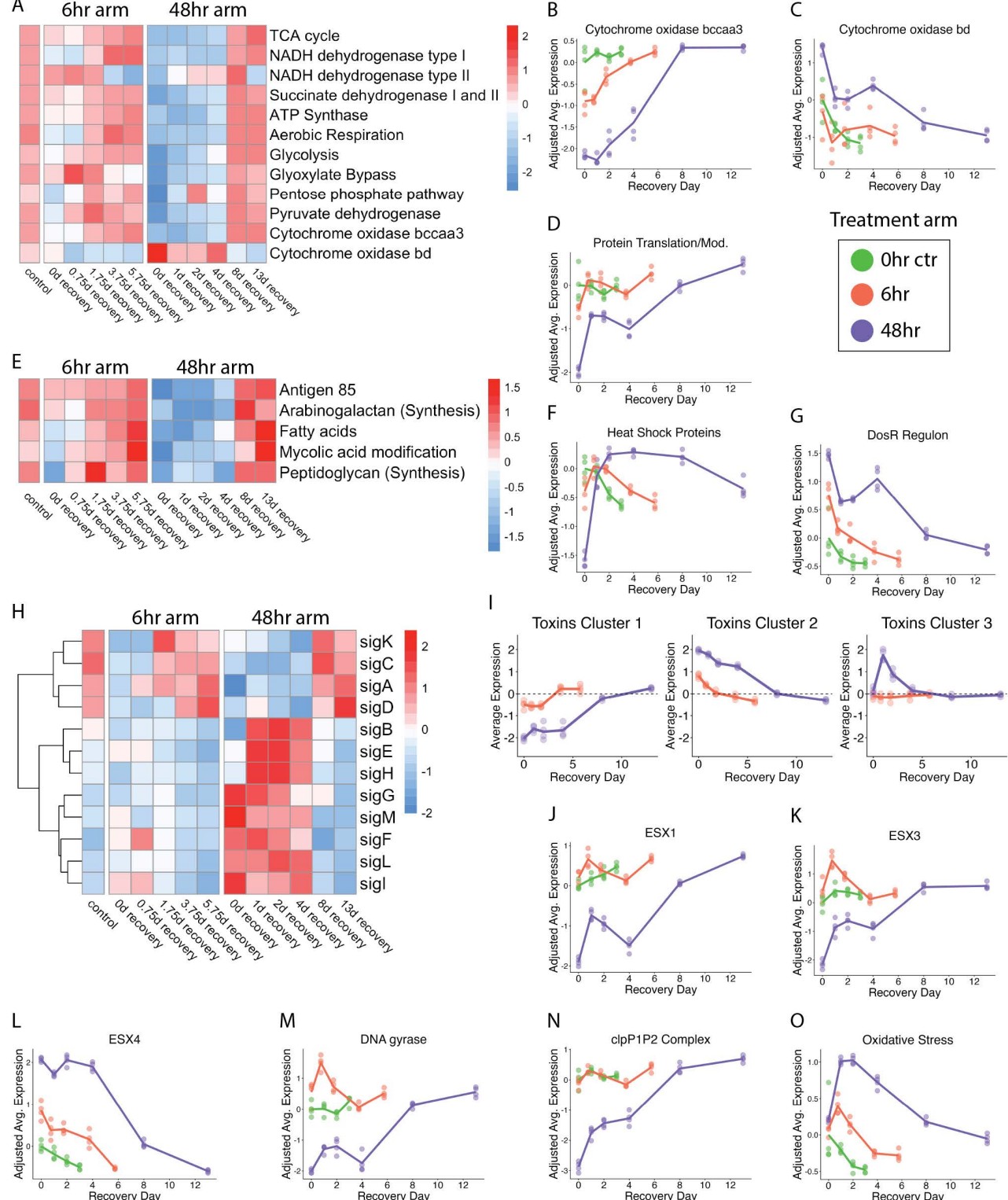

**Fig 3. Transcriptional recovery in axenic culture. (a)** The heatmap shows the batch-adjusted, VST-normalized, scaled gene expression averaged across samples for specified groups of gene categories related to metabolism. **(b-d)** Average of batch adjusted, VST-normalized, scaled gene expression in each treatment group over time for genes involved in (b) cytochrome *bcc/aa3* supercomplex, (c) cytochrome bd oxidase, and (d) protein

translation and modification. Each point represents an individual sample, and the lines connect the mean for each time point. Values are centered around the average value for the pre-treated samples so that points above and below zero represent upregulation and downregulation relative to pre-treated, respectively. **(e)** Heatmap as in (a) for gene categories relevant to cell wall synthesis and maintenance. **(f-g)** Average expression plots as in (b) for (f) chaperone heat shock proteins and (g) the DosR regulon. **(h)** Heatmap as in (a) for sigma factors A-M. **(i)** Hierarchical clustering of toxin genes (N = 73 genes) identified three broad patterns. Dot plots show the average expression for each cluster across time points. Each point represents an individual sample from pre-treatment control (black), 6-hour treatment (orange), or 48-hour treatment (purple). Horizontal lines indicate average values. Values are centered around the average value for the pre-treated samples so that points above and below zero represent upregulation and downregulation relative to pre-treatment, respectively. **(j-l)** Average expression plots as in (b) for **(j)** ESX1, **(k)** ESX3, **(l)** ESX4, **(m)** DNA gyrases, **(n)** the clpP1P2 complex, and **(o)** oxidative stress.

acid modification (*Adj-P<0.00001*), and peptidoglycan synthesis (*Adj-P<0.01*). An exception were fatty acid synthesis genes that decreased similarly in both arms. These changes were entirely reversed during recovery but recovery took longer in the 48-hour arm.

**Genes for chaperones and heat shock proteins *in vitro*.** Both 6-hour and 48-hour HRZE exposure decreased expression of genes for chaperones and heat shock proteins that facilitate protein folding under stress [36] (**Fig 3f**), but this effect was significantly greater for the 48-hour exposure (*Adj-P<0.00001*). For both the 6-hour and 48-hour arms, expression of chaperones rose immediately after drug exposure stopped, appearing among the first processes to recover.

**DosR regulon genes *in vitro*.** Both 6-hour and 48-hour HRZE exposure induced expression of the DosR regulon, that responds to impaired aerobic respiration (**Fig 3g**) [37–39], but this effect was significantly greater for the 48-hour exposure (*Adj-P<0.00001*). These changes were entirely reversed during recovery but recovery was slower in the 48-hour arm.

**Sigma factors genes *in vitro*.** HRZE altered expression of genes for sigma factors that are central regulators of *Mtb* gene expression(**Fig 3h**) [40]. Both 6-hour and 48-hour HRZE exposure decreased expression of *sigA*, encoding the primary growth-associated sigma factor, but this effect was significantly greater for the 48-hour exposure (*Adj-P<0.00001*). After removal of drug stress, expression of *sigA* gradually recovered to basline, albeit more slowly for the 48-hour arm than for the 6-hour arm. *SigC, sigD,* and *sigK* followed a pattern similar to *sigA*. By contrast, other sigma factors had relatively low expression prior to drug exposure, were induced by 48-hour HRZE exposure and resolved slowly during recovery.

**Toxin genes *in vitro*.** HRZE altered expression of genes for *Mtb*'s diverse repertoire of toxins, post-transcriptional stress response systems that regulate replication, translation, cell wall synthesis, and metabolism [41]. Hierarchical clustering identified three patterns of change (**Fig B in** S1 File). Cluster 1 comprised 14 toxin genes that were suppressed more strongly by 48-hour exposure than by 6-hour exposure. Expression of these toxin genes returned to baseline more slowly for the 48-hour arm (**Fig 3i**). Cluster 2 comprised 36 toxin genes that were induced more strongly by 48-hour exposure then by 6-hour exposure. Expression of these toxin genes returned to baseline more slowly for the 48-hour arm. Cluster 3 contained 23 toxin genes that rose at the start of recovery in the 48-hour arm only. Identifiers for the toxins in each cluster are summarized in **Table D in** S1 File.

**Genes coding for ESX loci *in vitro*.** HRZE affected expression of genes for the ESX Type VII secretion systems that are crucial for virulence, host interaction and nutrient uptake. Expression of genes in the ESX-1 locus which exports the strongly immunogenic ESAT-6 and CFP-10 proteins were not altered by 6-hour exposure but was suppressed by 48-hour exposure (*Adj-P<0.00001* for 6-hour vs 48-hour exposure; **Fig 3j**). Similarly, expression of ESX-3 genes which also modulate host response was not altered by 6-hour exposure but was suppressed by 48-hour exposure (*Adj-P<0.00001* for 6-hour vs 48-hour exposure; **Fig 3k**). By contrast, both 6-hour and 48-hour HRZE exposure induced expression of ESX-4, a transporter of unknown function that was previously shown to be induced by a variety of individual drugs [8,42] but this effect was greater after 48-hours (**Fig 3l**; *Adj-P<0.00001*). These changes reversed during recovery but recovery took longer for the 48-hour arm.

**DNA gyrase genes *in vitro*.** Expression of DNA gyrase genes necessary for DNA repair and replication increased significantly after 6-hour treatment (*Adj-P<0.001*) but decreased after 48-hour treatment (*Adj-P<0.00001*) (**Fig 3m**). For the 48-hour arm, DNA gyrase expression gradually recovered to control levels.

**ClpP1P2 protease complex genes *in vitro.*** Expression of genes for ClpP1P2 which degrades misfolded or damaged proteins was unaltered by 6-hour exposure but decreased after 48-hour exposure (*Adj-P<0.00001*) (Fig 3n). In the 48-hour arm, expression gradually returned to control levels.

**Oxidative stress response genes *in vitro.*** Expression of genes identified as responsive to hydrogen peroxide were not altered by either 6-hour or 48-hour exposure but did rise significantly at the start of recovery (*Adj-P<0.01*) at the first recovery timepoint relative to end of treatment for both 6-hour and 48-hour arms) (Fig 3o). With progressive recovery time, expression returned to control levels, more slowly for the 48-hour arm.

### Effect of HRZE treatment in mice

In mouse lungs, 2-week and 4-week HRZE treatment (Fig 4a) reduced CFU 97.2% (1.6 log reduction) and 99.0% (2.0 log reduction), respectively (Fig 4b). Before treatment, the median RS ratio was 204, indicating rapid rRNA synthesis. HRZE for 2-weeks and 4-weeks reduced the median RS ratio to 20.5 and 17.0 (Fig 4c), respectively, indicating substantially slowed rRNA synthesis. HRZE for 2-weeks and 4-weeks transformed the *Mtb* transcriptome in mice, significantly altering expression of 2,497 (70.3%) and 2,617 (73.7%) genes relative to pre-treatment control, respectively (**Fig C in** S1 File). These expression changes were broadly similar to changes previously reported with HRZE in mice (**Fig D in** S1 File) [8].

Consistent with our *in vitro* finding that longer treatment resulted in progressive transcriptional change, at end of treatment, the 4-week arm transcriptome was a slightly more extreme version of the 2-week transcriptome. Juxtaposing fold-changes relative to control after 2-weeks and 4-weeks showed a "shift" towards larger fold changes with longer treatment (Fig 4d) (albeit a shift of smaller magnitude than observed *in vitro* (Fig 1f)). The average absolute $\log_2$ fold change of significant genes was larger at 4-weeks (1.48) than at 2-weeks (1.35). In heirarchial clustering, Cluster 1 (N=1,259) contained genes suppressed by treatment, Cluster 2 (N=363) contained a small set of genes that were activated more by shorter treatment, and Cluster 3 (N=1,102) showed genes that were activated more after 4-week than 2-week treatment. Categorical enrichment of these clusters is included in **Table E in** S1 File.

### PK during the recovery period in mice

In the 4-week arm, we tested plasma drug concentrations one day after the final treatment dose. Pyrazinamide was detected at a trace (8.1 ng/mL) concentration [inhibitory concentration reported as 6,000–50,000 ng/mL at pH 5.5 [43], and > 64,000 ng/mL at pH 6.6, as tested here]. Ethambutol was detected at a sub-inhibitory (31.5 ng/ml) concentration [inhibitory concentration=1,000 ng/mL]. Rifampin was detected at an inhibitory (565.5 ng/mL) concentration [inhibitory concentration=8 ng/mL]. We additionally tested plasma drug concentrations five days after the final treatment dose and measured levels were below LOQ (Limit of Quantification) for of rifampin (LOQ-1 ng/mL) and pyrazinamide (LOQ-25 ng/mL). One of four mice tested had trace (1.1 ng/mL) concentration of isoniazid [inhibitory concentration=30 ng/mL]. All four mice tested had trace levels of ethambutol (2.49 ng/mL).

### Post-antibiotic effect in mice

When HRZE was stopped after 2 or 4 weeks, CFU counts stabilized and did not increase significantly during the subsequent 28-day post-antibiotic recovery period (Fig 5a; **Tables F-G in** S1 File). It was therefore not possible to calculate delay in time to CFU increase (the traditional measure of PAE) because recovery must require more than one month in this model.

### Physiologic recovery after drug exposure in mice

When HRZE was stopped after 2 weeks, the median RS ratio rebounded from 20.5 at end of treatment to 47.5 four days later, indicating resumption of rRNA synthesis (Fig 5b). Thereafter, the RS ratio increased more gradually, reaching 78.6 at the end of the 28-day post-antibiotic recovery period, a level characteristic of chronic immune-contained infection [20]. When HRZE was stopped after 4 weeks, the RS ratio recovered more gradually, with significantly lower values than in the 2-week group at all timepoints (least significant *Adj-P=0.049*; **Tables H-I in** S1 File).

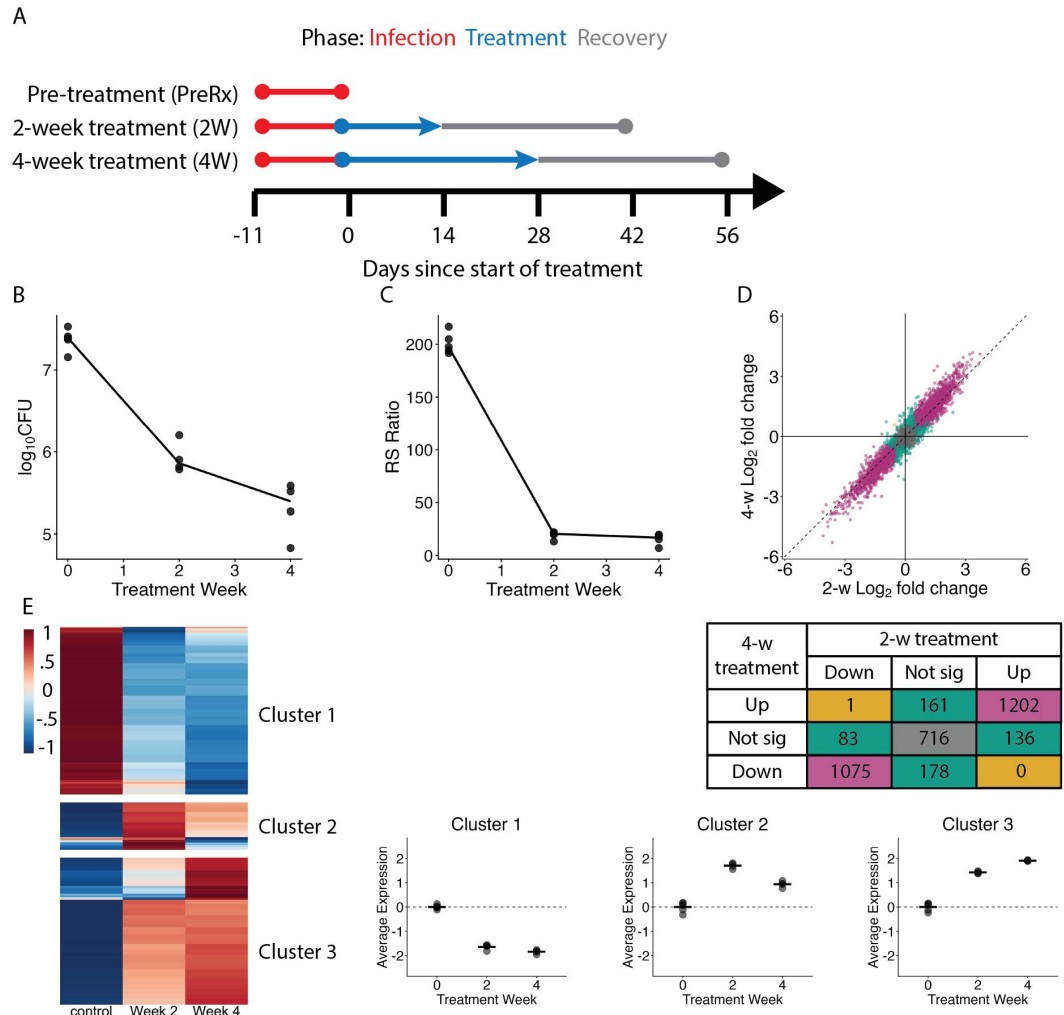

**Fig 4. Effect of treatment in murine samples. (a)** Timeline of mouse infection, treatment and PAE phase for experimental arms of murine study. Mice were untreated (preRx), treated for 2 weeks, or treated for 4 weeks. For treatment arms, treatment cessation was followed by a 28-day PAE phase. **(b)** CFU burden after 0, 2, and 4 weeks of treatment. Each dot represents one mouse sample. The line connects the median values. **(c)** RS ratio as displayed in **(b)**. **(d)** Log fold change of genes after 4-week treatment compared to pre-treatment control versus 2-week treatment compared to pre-treatment control. Purple shading indicates genes with concordant fold-change direction and significance between treatment times. Teal shading indicates genes that were significantly differentially expressed under one treatment duration but not both. Gold shading indicates genes that were significant for both treatment durations but in opposite directions. Gray shading indicates genes that were not significantly differentially expressed in either study. Table specifies the number of genes per category. **(e)** Hierarchical clustering of genes differentially expressed over time with HRZE treatment in murine samples (N = 2,724 samples). The heatmap shows the batch-adjusted, VST-normalized, scaled gene expression averaged across samples. Clustering identified three broad patterns. Dot plots show the average expression for each cluster across time. Each dot represents an individual sample. Horizontal lines indicate average values. Values are centered around the average value for the pre-treated samples so that points above and below zero represent upregulation and downregulation relative to pre-treatment, respectively.

The *Mtb* transcriptome recovered slowly (**Fig 5c**). Hierarchical clustering showed that little transcriptional change occurred during the first 14 days of recovery. For Clusters 2 (N = 953 genes) and 4 (N = 855), the transcriptome gradually trended back towards pre-treatment levels, similarly for the 2- and 4-week treatments. For Clusters 1 (N = 487) and 3 (N = 722), the 2-week arm transcriptome remained static for the entire 28-day recovery period while the 4-week transcriptome further diverged from pre-treatment phenotype. The rate of transcriptional recovery appeared slower for the 2-week arm (number of genes

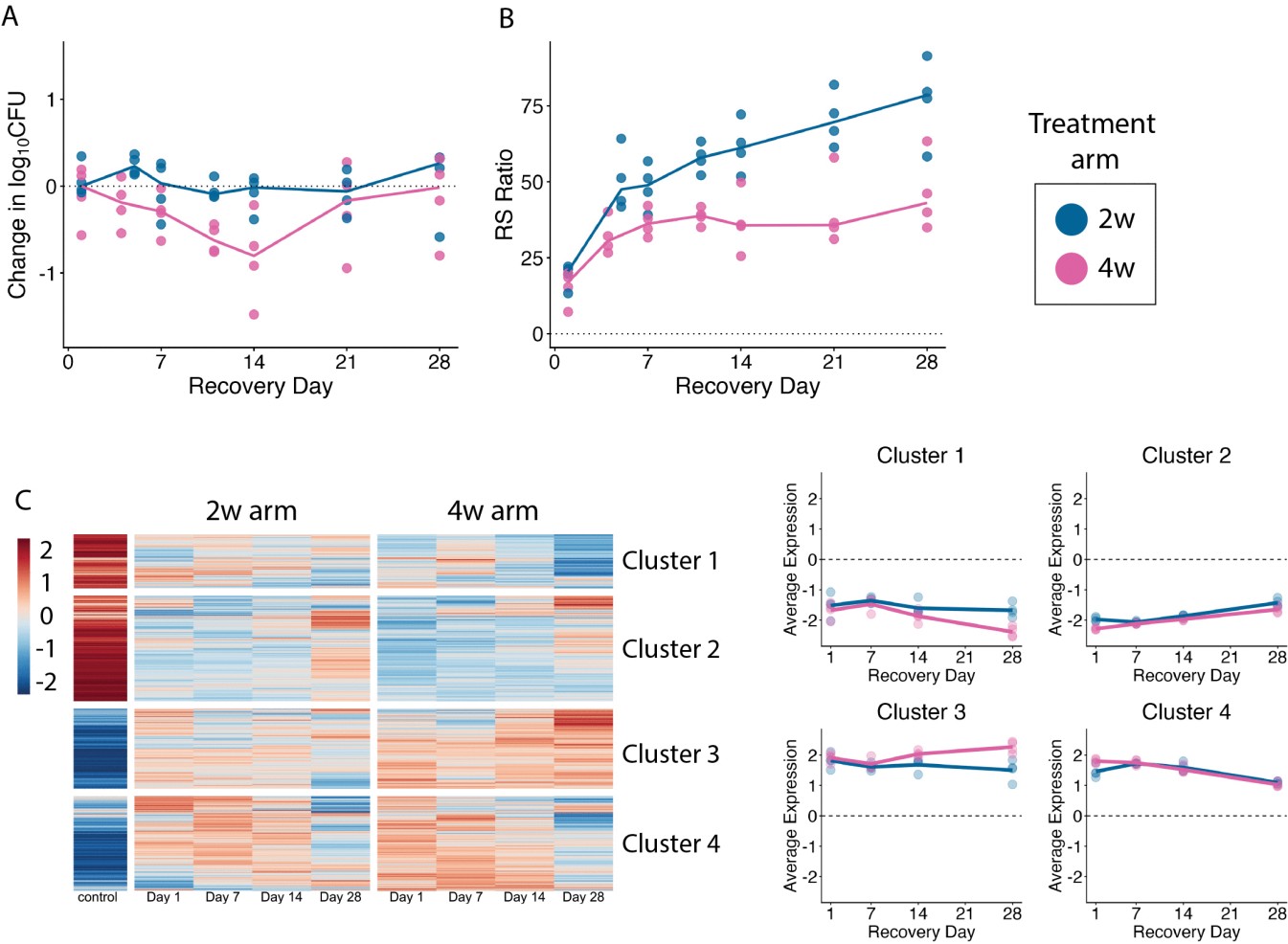

**Fig 5. Overview of recovery in murine samples. (a)** Change in CFU and **(b)** RS ratio of samples during the PAE phase of recovery. Each dot represents one sample, and lines connect the median values across timepoints for 2-week treatment (blue) and 4-week treatment (pink) groups. **(c)** Hierarchical clustering of genes differentially expressed over PAE phase (N = 3,017 genes). The heatmap shows the batch-adjusted, VST-normalized, scaled gene expression averaged across samples. Clustering identified four broad patterns. Dot plots show the average expression for each cluster across time points. Each point represents an individual sample from 2-week treatment (blue) or 4-week treatment (pink) group. Horizontal lines indicate average values. Values are centered around the average value for the pre-treated samples so that points above and below zero represent upregulation and downregulation relative to pre-treatment, respectively.

differentially expressed relative to end of treatment was 32, 150 and 800 on days 7, 14 and 28 post-treatment, respectively) than for the 4-week arm (16, 38 and 731 genes differentially expressed on days 7, 14 and 28, respectively).

## Comparison of specific *Mtb* physiologic processes in mice versus *in vitro* during recovery

Here, we summarize expression change during recovery in mice using change *in vitro* as a reference and highlighting differences observed in mice. As with the *in vitro* results, expression change can be further explored interactively via our Online Analysis Tool (https://microbialmetrics.org/analysis-tools/).

   **Metabolism-related genes in mice.** Concordant with *in vitro* observations (**Fig 3a**), HRZE reduced expression of metabolism-related genes in mice (**Fig 6a**). During recovery, expression of metabolism-related genes rose slowly. For

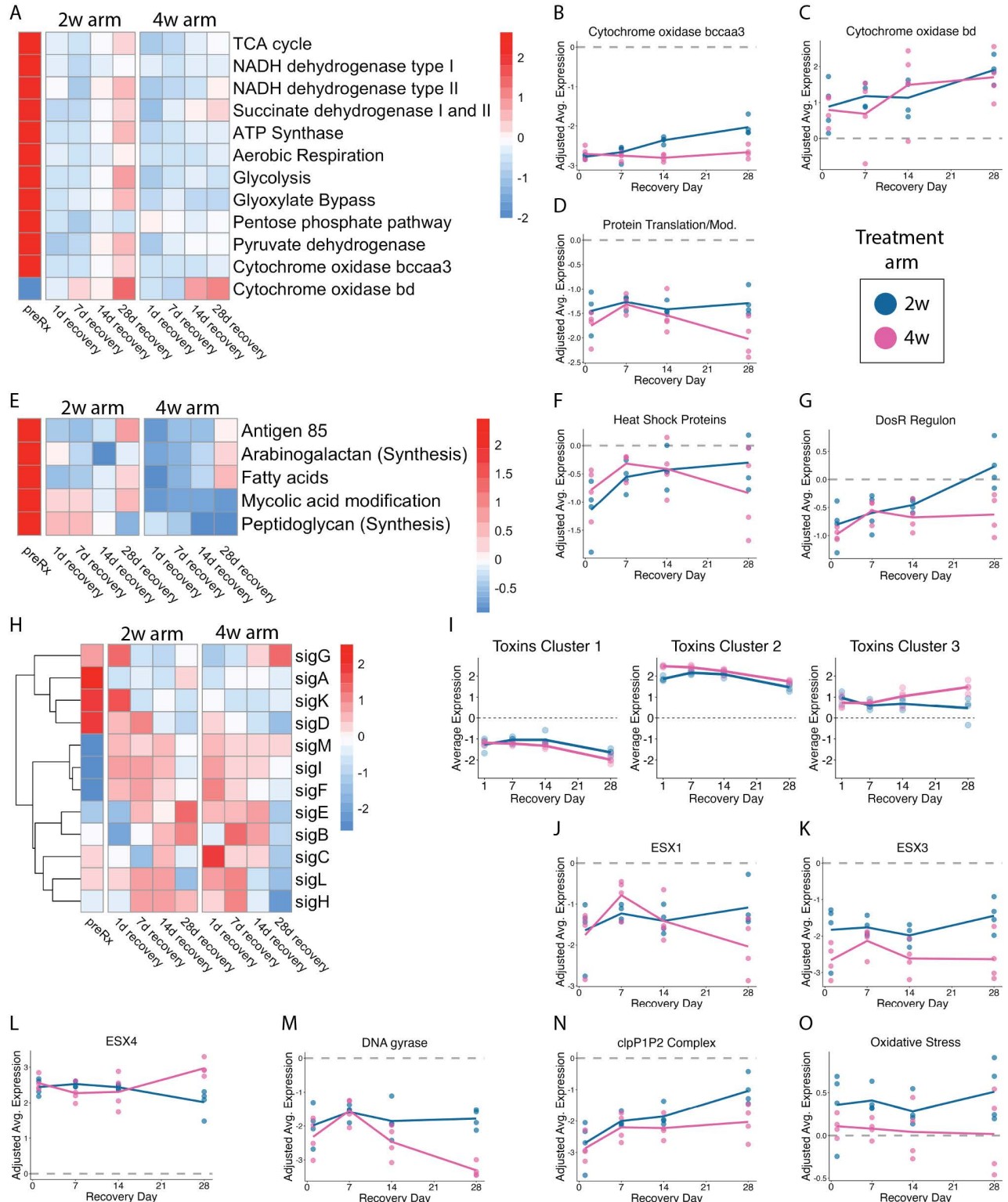

**Fig 6. Transcriptional recovery in murine samples. (a)** The heatmap shows the batch-adjusted, VST-normalized, scaled gene expression averaged across samples for specified groups of gene categories related to metabolism. **(b-d)** Average of batch adjusted, VST-normalized, scaled gene expression in each treatment group over time for genes involved in (b) cytochrome *bcc/aa3* supercomplex, (c) cytochrome bd oxidase, and (d) protein

translation and modification. Each point represents an individual sample, and the lines connect the mean for each time point. Values are centered around the average value for the pre-treated samples so that points above and below zero represent upregulation and downregulation relative to pre-treated, respectively. **(e)** Heatmap as in (a) for gene categories relevant to cell wall synthesis and maintenance. **(f-g)** Average expression plots as in (b) for (f) chaperone heat shock proteins and (g) the DosR regulon. **(h)** Heatmap as in (a) for sigma factors A-M. **(i)** Hierarchical clustering of toxin genes (N = 73 genes) identified three broad patterns. Dot plots show the average expression for each cluster across time points. Each point represents an individual sample from 2-week treatment (blue) or 4-week treatment (pink). Horizontal lines indicate average values. Values are centered around the average value for the pre-treated samples so that points above and below zero represent upregulation and downregulation relative to pre-treatment, respectively. **(j-l)** Average expression plots as in (b) for **(j)** ESX1, **(k)** ESX3, **(l)** ESX4, **(m)** DNA gyrases, **(n)** the clpP1P2 complex, and **(o)** oxidative stress.

several metabolic gene categories, the recovery was significantly greater among the 2-week arm relative to the 4-week arm. Specifically, ATP synthase (*Adj-P<0.01*), glycolysis (*Adj-P<0.00001*), glyoxylate bypass (*Adj-P<0.001*), and pyruvate dehydrogenase (*Adj-P<0.01*) genes. Unlike *in vitro*, *Mtb* in the murine samples never regained pre-treatment transcription levels within the observation period.

Change in expression of cytochrome oxidase genes in mice was similar to *in vitro* changes in certain respects and different in others. As observed *in vitro* (**Fig 3a**-**3c**), treatment in mice reduced expression of genes for the primary cytochrome *bcc/aa3* supercomplex (*Adj-P<0.00001* for 2-week and 4-week treatments) while there was a non-significant trend towards increased expression of the less-efficient alternative cytochrome *bd* oxidase (**Fig 6a**-**6c**). During recovery, expression of *bcc/aa3* genes gradually increased in the 2-week arm but did not approach pre-treatment levels. There was no appreciable change in *bcc/aa3* gene expression for the 4-week arm. In contrast to *in vitro* observations, expression of *bd* oxidase genes did not decline, in fact they displayed a trend towards increased expression suggesting persistent metabolic reprogramming in mice. Line plots for each gene set discussed above are included in Supplemental Information (**Fig E in** S1 File).

**Protein synthesis-related genes in mice.** Concordant with *in vitro* observations (**Fig 3d**; **Fig B in** S1 File), HRZE reduced expression of primary ribosomal protein genes (**Fig E in** S1 File; *Adj-P<0.00001* after 2-week and 4-week treatment) and protein translation and modification genes (**Fig 6d**; *Adj-P<0.00001* after 2-week and 4-week treatment) in mice. In contrast to the full recovery observed *in vitro*, neither the 2-week nor 4-week arms displayed significant recovery of protein synthesis-related genes. Nonetheless, at the end of the 28-day recovery period, expression of primary ribosomal protein and protein translation and modification genes was higher in the 2-week arm than the 4-week arm (*Adj-P<0.01* for both categories).

**Change in cell wall-associated genes in mice.** Concordant with *in vitro* observations (**Fig 3e**), HRZE reduced expression of genes related to *de novo* synthesis of cell wall constituents (**Fig 6e**). In contrast to the full recovery observed *in vitro*, few of the cell wall-associated categories rose significantly during the recovery period. The exceptions were Antigen 85 (*Adj=P<0.00001* for day 28 vs end of treatment for both 2-week and 4-week arms) and fatty acid synthesis (*Adj=P<0.012* for day 28 vs end of treatment for both 2-week and 4-week arms). No significance difference was observed between 2-week and 4-week arms and pre-treatment expression was not achieved (**Fig E in** S1 File).

**Genes for chaperones and heat shock proteins in mice.** Concordant with *in vitro* observations (**Fig 3f**), HRZE reduced expression of genes for chaperones and heat shock proteins. During recovery, expression of HSP genes rose for the 2-week arm (*Adj-P=0.033*) but not the 4-week arm (**Fig 6f**).

**DosR regulon genes in mice.** DosR regulon genes displayed discordant expression changes between *in vitro* (**Fig 3g**) and murine experiments (**Fig 6g**). In mice, HRZE suppressed DosR regulon expression (*Adj-P<0.01* after 2-week and 4-week treatment). During recovery, DosR expression rose in the 2-week arm (*Adj-P<0.01*) in parallel with the resumption of aerobic metabolism, eventually achieving expression levels comparable to pre-treatment expression. By contrast, DosR expression did not significantly recover in the 4-week arm, concordant with the slower metabolic recovery from longer treatment.

**Sigma factor genes in mice.** Concordant with *in vitro* observations (**Fig 3h**), HRZE reduced expression of the gene for the growth-enabling primary sigma factor SigA (*Adj-P=2.9x10^{-17}* for both 2-week and 4-week arms relative to control). During recovery, expression of *sigA* rose modestly but significantly (*Adj-P=0.046*) in the 2-week arm but did not rise in the 4-week arm (*Adj-P=0.6*). Other sigma factors were variably expressed during treatment and recovery (**Fig 6h**) similar to *in vitro*.

**Toxin genes in mice.** Concordant with *in vitro* observations (**Fig 3i**), HRZE markedly altered expression of toxin genes. Hierarchical clustering identified three broad categories (**Fig 6i**; Fig E in S1 File). Cluster 1 (N=21 genes) was suppressed by treatment and did not recover. Cluster 2 (N=38 genes) increased with treatment and decreased slightly during recovery. Cluster 3 (N=14 genes) was slightly activated with treatment and continued to rise in the 4-week arm but remained stagnant in the 2-week arm. None of these categories returned to pre-treatment expression levels by day 28. Identifiers for the toxins in each cluster are summarized in **Table J in** S1 File.

**Genes coding for ESX loci in mice.** Concordant with *in vitro* observations (**Fig 3j**-**3l**), HRZE suppressed expression of ESX-1 and ESX-3 (**Fig 6j**-**6k**). After exposure, the 2-week arm had a greater increase in ESX-1 (*Adj-P<0.05*) and ESX-3 (*Adj-P<0.001*) than the 4-week arm but nether arm fully recovered to control levels. Also concordant with *in vitro*, HRZE induced expression of ESX-4 (**Fig 6l**). After exposure, the 2-week arm had a greater decrease in expression of ESX-4 (*Adj-P<0.001*) than the 4-week arm but neither arm fully recovered to control levels.

**DNA gyrase genes in mice.** Concordant with the 48-hour *in vitro* arm (**Fig 3m**), HRZE reduced expression of DNA gyrase genes (**Fig 6m**). During recovery, expression of DNA gyrase genes did not increase yet remained significantly lower in the 4-week than in the 2-week arm at 28-days (Adj-P<0.001).

**ClpP1P2 protease complex genes in mice.** Concordant with the 48-hour *in vitro* arm (**Fig 3n**), HRZE reduced expression of ClpP1P2 genes (**Fig 6n**). During recovery, expression of ClpP1P2 genes did not increase yet was significantly higher in the 2-week arm relative to 4 weeks 28-days (Adj-P=0.014).

**Oxidative stress response genes in mice.** Unlike *in vitro* results in which oxidative stress response genes rose transiently during early recovery (**Fig 3o**), expression of these genes in mice was relatively static in mice, albeit trending higher in the 2-week arm than in the 4-week arm (Adj-P=0.056) (**Fig 6o**).

**Sustained differences between 2-week and 4-week arms.** During the recovery period, the transcriptional patterns of the 2-week and 4-week arms diverged rather than converged. Specifically, there were 229 genes differentially expressed between the 2-week and 4-week arms at the end of treatment. After the 28-day recovery period, the number of genes differentially expressed between 2-week and 4-week arms rose to 759.

## Discussion

Our evaluation of the physiologic recovery of *Mtb* during the PAE period *in vitro* and in mice identified three central findings. First, longer antibiotic exposure resulted in progressively greater bacterial injury and adaptation during drug exposure as well as slower recovery following exposure. Second, recovery followed a reproducible sequence—first initiation of rRNA synthesis, followed by broad transcriptional reprogramming, and finally increases in CFU. Third, while this sequence was conserved *in vitro* and in mice, bacterial recovery in mice was markedly slower and incomplete compared to *in vitro*, resulting in a substantially longer PAE. Together, these findings expand our understanding of bacterial recovery from antibiotic pressure and highlight key differences between recovery in controlled culture conditions and within a host.

PAE is classically defined as the sustained suppression of bacterial growth following transient antibiotic exposure (*i.e.,* after extracellular drug has been fully cleared).[44] In our *in vitro* experiments, extracellular drug was cleared via >1,000,000-fold dilution. In our mouse model, extracellular drug was cleared via metabolism or excretion such that all drugs were undetectable five days after treatment cessation (with the exception of trace isoniazid in a single mouse).

Engagement of antibiotics with their targets is known to initiate a downstream cascade of injury and adaptation [8,9,44–46] that disrupt essential processes [6,12]. If sustained, this cascade may reach a point where homeostasis can no longer be maintained—a concept that can be thought of as "death by depletion." These physiologic changes are not

measurable via CFU which only enumerates bacterial burden, providing no information about the state of bacterial physiologic processes. Similarly, CFU gives no information about bacterial physiology during the PAE period. We are not aware of previous evaluations of physiology as the bacterium recovers from drug stress.

Our *in vitro* and murine results showed that injury and adaptation are time-dependent. Measurment of RS ratio and SEARCH-TB *in vitro* demonstrated that drug stress resulted in rapid physiologic change that preceded change in bacterial burden. For example, 6 hours of HRZE had no effect on CFU but caused a > 10-fold reduction in the RS ratio (indicating interruption of rRNA synthesis) and significantly altered expression of >50% of *Mtb* genes (indicating a massive physiologic shift in a range of processes). Both *in vitro* and in mice, the magnitude of injury and adaptation depended on the duration of drug exposure, with transcriptional perturbations becoming progressively more extreme with time-on-treatment. This is consistent with sustained drug exposure driving a progressive cascade of injury and adaptation.

We also observed time-dependence in PAE. An association between longer exposure duration and longer PAE was described in early studies of streptococcus [47] and subsequently in other organisms [48], but to our knowledge has not previously been demonstrated for *Mtb* either *in vitro* or *in vivo*. Evidence of time-dependent PAE was pervasive and consistent in our experiments. Using the classical definition of PAE based on CFU change *in vitro,* we found that PAE was > 2.5-times longer after 48-hour exposure, relative to 6-hour exposure. *In vitro* and in mice, the RS ratio revealed that rRNA synthesis recovered more slowly following longer exposure. SEARCH-TB demonstrated *in vitro* and in mice that core physiological processes recovered more slowly after longer exposures.

A key observation in both *in vitro* and mouse experiments was sequential recovery of biologic processes. After the end of drug exposure, *Mtb* recovery began with resumption of rRNA synthesis, as indicated by a rapid increase in the RS ratio. Thereafter, there was a days-long pause before the emergence of meaningful transcriptional change. CFU was a lagging indiator, rising *in vitro* days after the transcriptome indicated recovery of metabolism and macromolecule synthesis. Although not previously reported in *Mtb* to our knowledge, a similar sequential order of recovery has been reported in *E. coli* and *B. subtilis* [49], starting with ribosome production before re-initiation of transcription of biosynthetic genes. The consistency of this finding from *in vitro* to mouse and across organisms suggests that there may exist a fundamental "order of operations" in the restoration of bacterial fitness.

Broadly, there are two non-exclusive hypotheses about why PAE occurs: 1) slow recovery of cellular function as bacteria repair nonlethal damage and (2) intracellular persistence of the drugs [48,50]. Although intracellular drug concentrations may remain elevated due to slow efflux or non-specific binding to cellular structures, the factor shown directly to influence PAE is drug-target residence time (i.e., duration an antibiotic remains bound before dissociating from its tartget). For example, long residence time is associated with the long PAE of ganfebrazole in *Mtb* [51] and has been demonstrated for rifampin, gentamycin and macrolides in other organisms [50,52] The importance of residence time in PAE is understood to depend on target vulnerability and the rate with which the target is synthesized and replenished [52]. Our current study was not designed to address the role of drug residence time in PAE.

Our results are most relevant to the nonlethal damage hypothesis. The time-dependent transcriptional change we observed during drug exposure is consistent with a progressive cascade of injury and adaptation. The time-dependent physiologic recovery we observed after drug exposure is consistent with slower recovery from "worse" injury. We note that long residence time alone would seem insufficient to explain the time-dependence of PAE we observed (*i.e.,*longer exposure = longer PAE) since drug dissociation is understood to be a constant that should not change with exposure duration [53] (*i.e.,* drug should not dissociate from its target more slowly after longer treatment than after shorter treatment). Critically, the nonlethal cellular damage hypothesis and long residence time hypothesis are not mutually exclusive. Both could contribute to PAE in combination.

We are not aware of previous measurement of the PAE of TB drugs *in vivo.* For other bacteria [21,22] it has been shown that PAE is substantially longer *in vivo* than *in vitro,* suggesting that immune stress may further delay bacterial recovery. Previous modeling has suggested that this would be the case for *Mtb* as well [23]. Our results validate these predictions. Whereas *Mtb* recovered over time to be nearly indistinguishable to the untreated control *in vitro, Mtb* in mice had only

"partial recovery" during the 4-week post-treatment period we monitored. *In vitro*, CFU, RS ratio and SEARCH-TB all indicated that *Mtb* returned to a rapidly replicating phenotype with a physiologic state similar to control. By contrast, in mice, CFU never rose significantly from end of treatment baseline, suggesting a slowly- or non-replicating state. The lack of rise in CFU meant PAE was non- quantifiable over a 28-day period using traditional measures, indicating very slow recovery. For mice, the RS ratio rose within days but only to a level previously observed in contained chronic infection. SEARCH-TB indicated recovery of some metabolic functions and a decrease in some stress responses but no discernable recovery in processes associated with macromolecule synthesis needed for growth. In contrast to the *in vitro* experiments in which *Mtb* in the 6-hour and 48-hour arms converged to be physiologically similar at the end of the post-exposure observation period, the 2-week and 4-week arms remained transcriptionally distinct at the end of the observation period. Collectively, these observations indicate that adaptive immunity plays a major role in limiting *Mtb* recovery even after antibiotics are cleared, whether through continued metabolic stress, restricted nutrient access, or other immune mechanisms.

Bacteria that tolerate drug exposure despite an absence of resistance-conferring mutations have been called "persisters" [54]. The small minority *Mtb* population that withstood weeks of HRZE treatment in mice (2.8% and 1.0% of the original population after 2-weeks and 4-weeks, respectively) could be considered persisters. However, this work and our previous work with individual drug exposures in mice [42] argue against a singular persister phenotype. Instead, we observe a time-dependent [8] and drug-specific [42] range of transcriptional injury and adaptation suggesting that there may be a spectrum of drug-tolerant states rather than a single a uniform persister phenotype.

This study had several limitations. We tested only the standard HRZE combination. Because individual drugs induce different patterns of physiologic injury and adaptation [8,10,55] and exhibit variable PAE [2,18,48,56] it seems likely that drugs may interact in ways that affect PAE. Molecular evaluation of additional regimens and individual components will be a next step. Second, we selected 28 days as our longest *in vivo* post-antibiotic timepoint with the expectation that *Mtb* would have fully recovered from drug exposure. It is conceivable that even longer post-treatment observations might have revealed further long-term transcriptional *Mtb* recovery. Third, as described above, findings in the mouse model were likely influenced by the onset of adaptive immunity. Repeating this study in an immunodeficient mouse model could isolate antibiotic recovery from the effects of the immune system.

Together, these results demonstrated that *Mtb* recovery from antibiotic stress is a multistage physiological process driven primarily by the extent of drug-induced cellular injury and the time required for cellular repair. Longer treatment imposed greater injury, slowed the restoration of ribosomal and transcriptional activity, and extended the lag before *Mtb* growth resumed. While complete recovery was achieved after transient exposure *in vitro*, recovery *in vivo* was slower, prolonged, and constrained by the host environment. Collectively, these findings support a damage–repair model of PAE, in which the duration of growth suppression reflects the time required to restore core cellular functions after progressive antibiotic-induced injury, and helps explain the durable bacteriostasis observed during treatment gaps or inconsistent adherence in TB therapy.

## Supporting information

**S1 File. Additional information on methods and results.**
(DOCX)

**S2 File. List of genes included within each *Mtb* gene category.**
(CSV)

## Author contributions

**Conceptualization:** Matthew J. Reichlen, Martin I. Voskuil, Gregory T. Robertson, Nicholas D. Walter.

**Data curation:** Reem Al Mubarak, Samuel T. Tabor, Lisa M. Massoudi, Karen Rossmassler, Firat Kaya.

**Formal analysis:** Jo Hendrix, Elizabeth A. Wynn, Camille M. Moore.

**Funding acquisition:** Gregory T. Robertson, Nicholas D. Walter.

**Investigation:** Jo Hendrix, Holly Nielsen.

**Methodology:** Reem Al Mubarak, Matthew J. Reichlen, Samuel T. Tabor, Adeline Bateman, Lisa M. Massoudi, Matthew D. Zimmerman, Holly Nielsen.

**Project administration:** Martin I. Voskuil, Gregory T. Robertson, Nicholas D. Walter.

**Resources:** Gregory T. Robertson, Nicholas D. Walter.

**Supervision:** Gregory T. Robertson, Nicholas D. Walter.

**Visualization:** Jo Hendrix.

**Writing – original draft:** Jo Hendrix, Nicholas D. Walter.

**Writing – review & editing:** Jo Hendrix, Reem Al Mubarak, Matthew J. Reichlen, Samuel T. Tabor, Adeline Bateman, Lisa M. Massoudi, Karen Rossmassler, Firat Kaya, Matthew D. Zimmerman, Holly Nielsen, Elizabeth A. Wynn, Martin I. Voskuil, Gregory T. Robertson, Camille M. Moore, Nicholas D. Walter.

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
