## [Decision Letter · Decision Letter 0]

25 Jun 2025

Physiologic recovery of Mycobacterium tuberculosis from drug injury: A molecular study of post antibiotic effect in mice

PLOS Pathogens

Dear Dr. Walter,

Thank you for submitting your manuscript to PLOS Pathogens. After careful consideration, we feel that it has merit but does not fully meet PLOS Pathogens's publication criteria as it currently stands. Therefore, we invite you to submit a revised version of the manuscript that addresses the points raised during the review process.

Please submit your revised manuscript within 60 days Aug 24 2025 11:59PM. If you will need more time than this to complete your revisions, please reply to this message or contact the journal office at plospathogens@plos.org. Please include the following items when submitting your revised manuscript:

We look forward to receiving your revised manuscript.

Kind regards,

David M. Lewinsohn

Academic Editor

PLOS Pathogens

Michael Otto

Section Editor

PLOS Pathogens

Editor-in-Chief

PLOS Pathogens

Editor-in-Chief

PLOS Pathogens

orcid.org/0000-0002-7699-2064

**Additional Editor Comments :**

We thank the authors for their patience, and find the manuscript well received. Please address the in vitro concern of reviewer #1, ideally experimentally.

**Journal Requirements:**

- ® on pages: 2, and 5.

Potential Copyright Issues:

i) Figure 1b. Please confirm whether you drew the images / clip-art within the figure panels by hand. If you did not draw the images, please provide (a) a link to the source of the images or icons and their license / terms of use; or (b) written permission from the copyright holder to publish the images or icons under our CC BY 4.0 license. Alternatively, you may replace the images with open source alternatives. See these open source resources you may use to replace images / clip-art:

5) Thank you for stating "Sequence data are available via accession number PRJNA1140268." Please note that, though access restrictions are acceptable now, your entire minimal dataset will need to be made freely accessible if your manuscript is accepted for publication. This policy applies to all data except where public deposition would breach compliance with the protocol approved by your research ethics board. If you are unable to adhere to our open data policy, please kindly revise your statement to explain your reasoning and we will seek the editor's input on an exemption.

2) If any authors received a salary from any of your funders, please state which authors and which funders.

7) Please ensure that the funders and grant numbers match between the Financial Disclosure field and the Funding Information tab in your submission form. Note that the funders must be provided in the same order in both places as well. Currently, the order of the grants is different in both places.

8) Please provide a completed 'Competing Interests' statement, including any COIs declared by your co-authors. If you have no competing interests to declare, please state "The authors have declared that no competing interests exist".

**Reviewers' Comments:**

Reviewer's Responses to Questions

**Part I - Summary**

Reviewer #1: The post-antibiotic effect (PAE) is a critical component of in vivo drug activity for many antibiotics.

For rapid growing Gram positive and Gram negative bacteria, antibiotics can be classified based on whether or not they generate a PAE which can then be explicitly linked to the dosing regimen – i.e. drugs that generate a PAE are dosed less frequently.

While the in vitro PAE has been measured for TB drugs, there are no specific studies that quantify in vivo PAE. Consequently, the work described in the manuscript is important and timely. Both the novelty and significance are very high!

The use of the RS ratio to look at the recovery of rRNA synthesis compared to overall transcriptional changes is a major strength of the work since this enables the authors to start to draw conclusions regarding the mechanism(s) that underlie the observed data. However this should be applied to in vitro PAE studies to provide a better connection between in vitro and in vivo experiments. This would then provide a framework for dissecting the contributions of the individual antibiotics to the PAE.

Clearly measuring in vivo PAEs for individual drugs and different combinations would be a herculean task – the reviewer isn’t suggesting that this be done. However, as the authors point out, in vitro PAEs are normally measured by CFU counting. Thus a link between rRNA synthesis, physiological changes assessed by transcriptional response and CFU counting for in vitro PAEs of one or more drugs plus the combination would substantially strengthen the work.

Reviewer #2: The authors present an interesting manuscript that addresses post-antibiotic effect in TB-infected mice treated with standard anti-TB therapy. In general the presentation is clear and the data are thought-provoking. I was left with a few questions and thoughts:

• RS ratios could be described a little more in methods and results/discussion. How are they determined and what do they actually mean? The authors refer back to their earlier work, but I could not find enough detail to think I’d be able to reproduce that work.

• The week 2 and week 4 treatment groups are compared throughout the manuscript but there are no statistical analyses comparing then, can statistics be performed between these groups? For example, adding statistics to Fig 3b-l.

• It would be good to see the variation between mouse replicates within each treatment group. How does that variation compare to the differences between week 2 and week 4? Can differences between week 2 and week 4 groups be explained by variation between mouse replicates?

• Line 214: Acronyms BLQ and LOQ are used but not described.

• Can the authors better define the expected transcriptome of bacteria experiencing post-antibiotic effect (PAE) and if their results match what is expected? We might for example imagine that proteins and/or DNA become damaged and need repair. Are proteases/DNA repair enzymes/oxidative stress response genes differentially regulated?

• Can the authors include a supplemental file of all their curated gene sets and the list of genes in each set? This would help other researchers!

• Fig S2: Can the authors explain why there are significant differences between the two experiments, and why the relationship is not linear with the more highly expressed genes?

• The authors structure their manuscript around “treatment forgiveness” and the physiological changes that happen to bacteria in vivo when drug treatment is stopped. However, they do not present a transcriptome from Mtb that recover after treatment interruption or show how long the PAE lasts. It may be better to decrease focus on treatment forgiveness.

**Part II – Major Issues: Key Experiments Required for Acceptance**

Please use this section to detail the key new experiments or modifications of existing experiments that should be absolutely required to validate study conclusions.required to validate study conclusions.required to validate study conclusions.required to validate study conclusions.

Reviewer #1: The reviewer recommends that the authors perform in vitro PAEs using RS ratio and transcriptional profiling in addition to CFU counting to provide a better link between in vitro and in vivo studies. This would provide additional insight into the mechanism(s) of the PAE. An experiment with the HRZE combination would be information that of course studies with individual drugs would be much better.

Reviewer #2: None

**Part III – Minor Issues: Editorial and Data Presentation Modifications**

Reviewer #1: Below are some comments and questions that should be addressed prior to publication.

(i) Additional background could be provided regarding the mechanism and importance of the PAE. For instance, that generation of a PAE is linked to dosing regimens (albeit for rapid growing bugs). In addition, there are actually several proposed mechanisms for the PAE that also include sequestration of antibiotic by the bacterium and that the antibiotic is still bound to the target. For instance, the relationship between the lifetime of the drug-target complex has been linked to target vulnerability. The authors could expand on proposed mechanisms and add appropriate references.

(ii) The investigators use a well-established drug cocktail for their experiments. Whilst this is relevant to the treatment of human disease, it would be interesting to know whether one or more of the drugs in the cocktail have a major effect on PAE. The authors state that studies of the in vivo PAE are more relevant than in vitro studies, however it would be useful to include in vitro data PAEs. For instance rifampin is known to generate a long in vitro PAE (70 h) whereas isoniazid generates a shorter PAE (18 h) and ethambutol a very short PAE (2 h). In contrast H+R is ~160 h.

(iii) The observation that rRNA synthesis occurs well before the transcription of many other genes is a highly interesting observation. It would be interesting to know mechanistically why this is – is rRNA synthesis separate from other gene transcription events?

(iv) In addition what is the relationship between the physiological damage observed by transcriptional profiling and bacterial growth? This seems to be a critical gap in knowledge – e.g. how much ‘repair’ is needed before regrowth can commence?

(v) Some might argue that the bacteria that regrow following drug exposure represent a ‘persister’ population. The authors might like to add some discussion regarding this point.

Reviewer #2: The authors present an interesting manuscript that addresses post-antibiotic effect in TB-infected mice treated with standard anti-TB therapy. In general the presentation is clear and the data are thought-provoking. I was left with a few questions and thoughts:

• RS ratios could be described a little more in methods and results/discussion. How are they determined and what do they actually mean? The authors refer back to their earlier work, but I could not find enough detail to think I’d be able to reproduce that work.

• The week 2 and week 4 treatment groups are compared throughout the manuscript but there are no statistical analyses comparing then, can statistics be performed between these groups? For example, adding statistics to Fig 3b-l.

• It would be good to see the variation between mouse replicates within each treatment group. How does that variation compare to the differences between week 2 and week 4? Can differences between week 2 and week 4 groups be explained by variation between mouse replicates?

• Line 214: Acronyms BLQ and LOQ are used but not described.

• Can the authors better define the expected transcriptome of bacteria experiencing post-antibiotic effect (PAE) and if their results match what is expected? We might for example imagine that proteins and/or DNA become damaged and need repair. Are proteases/DNA repair enzymes/oxidative stress response genes differentially regulated?

• Can the authors include a supplemental file of all their curated gene sets and the list of genes in each set? This would help other researchers!

• Fig S2: Can the authors explain why there are significant differences between the two experiments, and why the relationship is not linear with the more highly expressed genes?

• The authors structure their manuscript around “treatment forgiveness” and the physiological changes that happen to bacteria in vivo when drug treatment is stopped. However, they do not present a transcriptome from Mtb that recover after treatment interruption or show how long the PAE lasts. It may be better to decrease focus on treatment forgiveness.

PLOS authors have the option to publish the peer review history of their article (what does this mean?). If published, this will include your full peer review and any attached files.). If published, this will include your full peer review and any attached files.). If published, this will include your full peer review and any attached files.). If published, this will include your full peer review and any attached files.

...

Reviewer #1: No

Reviewer #2: No

**Figure resubmission:**

**Reproducibility:**



---

## [Decision Letter · Decision Letter 1]

21 Jan 2026

PPATHOGENS-D-25-00711R1

Physiologic recovery of Mycobacterium tuberculosis from drug injury:

A molecular study of post antibiotic effect in vitro and in vivo

PLOS Pathogens

Dear Dr. Walter,

Thank you for submitting your manuscript to PLOS Pathogens. After careful consideration, we feel that it has merit but does not fully meet PLOS Pathogens's publication criteria as it currently stands. Therefore, we invite you to submit a revised version of the manuscript that addresses the points raised during the review process.

We look forward to receiving your revised manuscript.

Kind regards,

David M. Lewinsohn

Academic Editor

PLOS Pathogens

Michael Otto

Section Editor

PLOS Pathogens

Sumita Bhaduri-McIntosh

Editor-in-Chief

PLOS Pathogens

orcid.org/0000-0003-2946-9497

Michael Malim

Editor-in-Chief

PLOS Pathogens

orcid.org/0000-0002-7699-2064

**Journal Requirements:**

1)  Please ensure that the funders and grant numbers match between the Financial Disclosure field and the Funding Information tab in your submission form. Note that the funders must be provided in the same order in both places as well.

**Reviewers' Comments:**

Reviewer's Responses to Questions

**Part I - Summary**

Reviewer #1: The authors have significantly improved the manuscript and I acknowledge the large amount of extra work that has gone into the revision!

There are a couple of details the authors might consider.

1) Regarding this point:

There are two general hypotheses about why PAE occurs: 1) slow recovery of cellular function as bacteria repair nonlethal damage and (2) intracellular persistence of the drugs (i.e., non-dissociation from target).48,50

Intracellular persistence of drug is normally used to refer to a general scenario where the drug remains in the cell. This can occur through different mechanisms. For instance, the drug could be trapped in the cell (slow efflux) or nonspecifically bound to cellular structures. In my review I was referring to a specific mechanism in which the drug is still bound to the drug target because the drug has a long residence time on the target.

In all these scenarios the drug will remain bound to the target when the extracellular drug concentration has decreased (e.g. in the washout step of a PAE). However, only in the situation where the drug has a long residence time on the target (slow off rate or covalent) will non-dissociation from the target actually be the reason for drug still being in the cell which as implied by the authors.

I think this is an important distinction between the general phenomenon of intracellular persistence of the drug and the lifetime of the drug target complex, which I encourage the authors to make in their manuscript and add appropriate references. For instance, there are examples where the PAE has been explicitly linked to the slow off rate of the drug from the target. This is the proposed mechanism for the long PAE of ganfeborole in Mtb (doi:10.1021/acschembio.5c00705). There are also other examples – e.g. the ribosome and LpxC).

Of course, whether or not long drug-target residence time translates to a PAE depends on target vulnerability and the rate of target synthesis (e.g. covalent b-lactam inhibitors of PBPs do not generate a PAE in Gram negatives). This was the other point I was making in my review.

2) Secondly, there are a number of articles that refer to the observation that in vivo PAEs are often longer than in vitro PAES – this could be due to immune pressure, nutrient limitation that alters target vulnerability or possibly trapping/accumulation of drug in the infected tissue. Thus, I am not sure that it is warranted to use the word ‘novel’ in this context:

…novel finding that PAE is longer in mice than in vitro suggests that immunity may augment PAE

I acknowledge that most in vivo studies are for rapid growing bacteria. It is of course very difficult to know what the local drug concentration is in the infected tissue but it is possible that the tissue acts as a reservoir that clears more slowly. PET imaging or ex vivo mass spec imaging are ways to do this (albeit non trivial).

**Part II – Major Issues: Key Experiments Required for Acceptance**

Please use this section to detail the key new experiments or modifications of existing experiments that should be absolutely required to validate study conclusions.required to validate study conclusions.required to validate study conclusions.required to validate study conclusions.

Reviewer #1: (No Response)

**Part III – Minor Issues: Editorial and Data Presentation Modifications**

Reviewer #1: I am only requesting that the authors consider minor edits to address points 1 and 2 above.

PLOS authors have the option to publish the peer review history of their article (what does this mean?). If published, this will include your full peer review and any attached files.). If published, this will include your full peer review and any attached files.). If published, this will include your full peer review and any attached files.). If published, this will include your full peer review and any attached files.

...

Reviewer #1: **Yes:** Peter TongePeter TongePeter TongePeter Tonge

**Figure resubmission:**
---

## [Editor Report · Decision Letter 2]

30 Mar 2026

Dear Dr. Walter,

We are pleased to inform you that your manuscript 'Physiologic recovery of Mycobacterium tuberculosis from drug injury:

A molecular study of post antibiotic effect in vitro and in vivo' has been provisionally accepted for publication in PLOS Pathogens.

Best regards,

David M. Lewinsohn

Academic Editor

PLOS Pathogens

Michael Otto

Section Editor

PLOS Pathogens

Sumita Bhaduri-McIntosh

Editor-in-Chief

PLOS Pathogens

orcid.org/0000-0003-2946-9497

Michael Malim

Editor-in-Chief

PLOS Pathogens

orcid.org/0000-0002-7699-2064
---

## [Editor Report · Acceptance letter]

Dear Dr. Walter,

We are delighted to inform you that your manuscript, "Physiologic recovery of Mycobacterium tuberculosis from drug injury:

A molecular study of post antibiotic effect in vitro and in vivo," has been formally accepted for publication in PLOS Pathogens.

Best regards,

Sumita Bhaduri-McIntosh

Editor-in-Chief

PLOS Pathogens

orcid.org/0000-0003-2946-9497

Michael Malim

Editor-in-Chief

PLOS Pathogens

orcid.org/0000-0002-7699-2064